# Fluorinated Ethylene Propylene Coatings Deposited by a Spray Process: Mechanical Properties, Scratch and Wear Behavior

**DOI:** 10.3390/polym14020347

**Published:** 2022-01-17

**Authors:** Najoua Barhoumi, Kaouther Khlifi, Abderrahim Maazouz, Khalid Lamnawar

**Affiliations:** 1Laboratoire de Mécanique, Matériaux et Procédés, Ecole Nationale Supérieure d’Ingénieurs de Tunis, Université de Tunis, 5, Avenue Taha Husseïn, Montfleury, Tunis 1008, Tunisia; kaouther.khlifi@ipeiem.utm.tn; 2Institut Préparatoire aux Etudes d’Ingénieurs d’El-Manar, Université d’El-Manar, B.P 244, Tunis 2092, Tunisia; 3CNRS, UMR 5223, Ingénierie des Matériaux Polymères, INSA Lyon, Université de Lyon, F-69621 Villeurbanne, France; abderrahim.maazouz@insa-lyon.fr

**Keywords:** spray process, FEP coating, scratch behavior, friction and wear resistance

## Abstract

To increase the lifetime of metallic molds and protect their surface from wear, a fluorinated ethylene propylene (FEP) polymer was coated onto a stainless-steel (SS304) substrate, using an air spray process followed by a heat treatment. The microstructural properties of the coating were studied using scanning electron microscopy (SEM) and energy-dispersive X-ray spectroscopy (EDS) as well as X-ray diffraction. The mechanical properties and adhesion behavior were analyzed via a nanoindentation test and progressive scratching. According to the results, the FEP coating had a smooth and dense microstructure. The mechanical properties of the coatings, i.e., the hardness and Young’s modulus, were 57 ± 2.35 and 1.56 ± 0.07 GPa, respectively. During scratching, successive delamination stages (initiation, expansion, and propagation) were noticed, and the measured critical loads L_C1_ (3.36 N), L_C2_ (6.2 N), and L_C3_ (7.6 N) indicated a high adhesion of the FEP coating to SS304. The detailed wear behavior and related damage mechanisms of the FEP coating were investigated employing a multi-pass scratch test and SEM in various sliding conditions. It was found that the wear volume increased with an increase in applied load and sliding velocity. Moreover, the FEP coating revealed a low friction coefficient (around 0.13) and a low wear coefficient (3.1 × 10^−4^ mm^3^ N m^−1^). The investigation of the damage mechanisms of the FEP coating showed a viscoelastic plastic deformation related to FEP ductility. Finally, the coating’s resistance to corrosion was examined using electrochemical measurements in a 3.5 wt% NaCl solution. The coating was found to provide satisfactory corrosion protection to the SS304 substrate, as no corrosion was observed after 60 days of immersion.

## 1. Introduction

Stainless steel is widely use in the fabrication of molds elements using in the chemical and food processing industry, in metal forming and pharmaceutical molds [1,2,3,4]. In sliding mechanical systems during service, wear and friction of these metallic components produce several problems and lead to scratches, severe wear, and plastic deformation [5]. Furthermore, in the strongly aggressive media containing chloride anions, stainless steels lack corrosion resistance [6,7]. These failure mechanisms directly involve the molds’ surface, creating a serious problem in the manufacturing of casting, and shortens molds’ life [8,9]. To solve this problem, which causes additional costs for mold reconditioning and reduced productivity, various metallic, ceramic, and polymeric coatings have been proposed. Nickel-based metallic and zirconia-based ceramic protective coatings were deposited on steel diecasting molds [8]. According to Óscar Rodríguez-Alabanda [9], fluoropolymer coatings can be applied as anti-adherent coatings on aluminum–magnesium substrates for food containers. In our previous work [10], polymeric perfluoroalkoxy coatings were used to protect agri-food molds to prevent corrosion and wear.

Fluorinated polymeric coatings such as polytetrafluoroethylene (PTFE), perfluoroalkoxy (PFA) and fluorinated ethylene propylene (FEP) are the frequently used fluoroplastics [11,12,13]. These fluorinated polymers are widely applied on molds elements used in the food sector to prevent adherence to substrates, to achieve low chemical reactivity and good corrosion resistance, to facilitate cleaning, to improve unmolding, and to increase the service life of mold elements by protecting their surface from cavity damage through abrasive wear [14]. In this way, chemical stability at high and low temperatures is obtained, and the coating process is simple, short, and leads to low friction coefficients [6,14,15,16]. These outstanding properties of anti-adherent fluoropolymers result from the fact that C–C bonds are strengthened by the incorporation of fluorine atoms into organic materials. In addition, the protective properties of fluoropolymer coatings can be related to the chemical (fluorine and chlorine functional groups) and physical (significant ordering of the polymeric chains) characteristics of the resins [17].

Thanks to its self-lubricating properties, FEP is extensively used in polymer/metal tribological systems working in dry, mixing, and wet environments [18,19]; it also induces high corrosion resistance due to its action as a barrier layer between corrosive species and the metal surface [20]. Fluorinated polymers can be produced by two processes. One of them involves the spray of water-based dispersions of fluoropolymer resins, followed by heat treatment at a temperature between 260 °C and 360 °C. The thickness of the smooth film produced by this method can reach 1 mm [21]. The second process involves the electrostatic spraying of polymer powders on a substrate [22]. The first process (spray) is a simple spraying technique with growing scientific interests and industrial applications in the field of industrial molds and components. This method offers corrosion protection and results in increases in mechanical and physical durability and wear resistance.

Recently, efforts have been devoted to exploring the scratch and wear behavior and to enhancing the corrosion and wear resistance of different forms of steel by fluoropolymer coating deposition [8,9,15,16]. The coating adhesion of FEP to SS304 is known to be critical, and many studies have been carried out to investigate the scratch behavior and related damage mechanisms of polymeric materials [23] and the adhesion behavior of coatings [24,25]. Xu et al. [23] studied the scratch behavior of alternating multi-layered PMMA/PC materials and found that the delamination process during scratching could be divided into three stages of delamination (initiation, expansion, and propagation).

The protective properties of an FEP coating in saline aqueous solutions, its adhesion using scratch tests, and its wear behavior using multi-pass scratch tests have not yet been investigated. The present work aimed to develop non-stick coatings to protect the SS304 mold substrate and to ameliorate its surface performance. Stainless-steel specimens (SS304) coated with the fluoropolymer FEP were prepared using the spray coating technique. The process parameters of FEP deposit are illustrated. The structure of the FEP coating and its adhesion were studied. Nanoindentation tests were used to investigate the mechanical properties, and the friction and wear properties were explored using multi-pass scratch testing. The aim of this work was to develop FEP-based coatings, sprayed on SS304 steel, and to propose them as a suitable alternative in the manufacture of wear-resistant, anti-adherent, and corrosion-resistive molds.

## 2. Materials and Methods

### 2.1. Material

A stainless steel (SS304) sheet with a chemical composition according to Table 1 was machined to rectangular specimens with dimensions of 100 × 80 × 2 (mm^3^). The FEP resin, supplied by Whitford (England) as an aqueous solution with a purity of 99.9%, was used to coat the steel substrates. The coating deposition was carried out by RIET industry (Zaghouan—TUNISIa).

### 2.2. Coating Deposition

Before coating deposition, the steel substrate was cleaned with acetone and polished with a series of silicon carbide sandpapers. Subsequently, it was roughened by sandblasting to a surface finish Ra ≈ 3 µm. Then, it was dried at 60 °C for 24 h. Figure 1 details the deposition of the FEP solution onto the substrate surface using the air spray process. An FEP coating was obtained by repeating the spraying–drying process twice. The spray conditions were maintained at a liquid flow rate of 2.5 mL min^−1^, an air pressure of 3 bars, and a nozzle-to-substrate distance of 8 cm.

The FEP solution was first applied by air gun spraying to the substrate, after which the materials were dried at 150 °C for 20 min, as shown in Figure 1. This was followed by a second round of the specimens being spray-coated with FEP and dried in an oven at 150 °C for 20 min. Subsequently, they were cured at 380 °C for 25 min for the coating to melt and spread over the substrate. Finally, they were cooled to ambient temperature. The thickness of the coatings was measured with a standard gauge (Ecotest plus device) and was found to be in the range of 44 ± 4 µm. Mechanical and wear characterization tests were conducted and repeated five times on the deposited FEP coatings.

### 2.3. Structural Characterization

The film composition and the microstructure of FEP coatings were evaluated using Scanning Electron Microscopy (SEM-JEOL JSM 6460LV -Suisse) in combination with energy-dispersive X-ray spectroscopy (EDS) for the identification of the chemical composition. X-ray spectra were obtained at a primary beam energy equal to 20 keV with an acquisition time of 120 s. The study of the phase composition of the samples was performed using X-ray diffraction (Bruker D8 Advance), and X-ray patterns were gathered from 2θ = 10° to 70°.

### 2.4. Nanoindentation Testing

The mechanical properties of the FEP coatings (hardness and Young’s modulus) were studied using the nanoindentation technique developed by CSM Instruments (Anton Paar, Graz, Austria). Tests were conducted with a nanoindenter equipped with a diamond Berkovich tip with a nominal angle equal to 65.3° and a nominal radius curvature of 20 nm. The maximum applied load was 10 mN, and the penetration depth was set to a value lower than 1/10th of the coating thickness to provide the real film properties and avoid the substrate effect [26]. A minimum of 15 measurements were carried out for each coating to ensure the reproducibility and repeatability of the results. The analysis of load–displacement graphs with the Oliver and Pharr method [27] allowed an estimation of the mechanical properties (Young’s modulus, hardness of the film).

### 2.5. Scratch Testing

Film adhesion was investigated with a micro-scratch tester from CSM Instruments (Anton Paar). The indenter was equipped with a Rockwell diamond tip with a radius of 200 μm. The test was performed in progressive mode in a loading range between 0.3 and 25 N along 3 mm of the scratch length, to study the adhesion behavior and to determine the critical loads of the film. The scratch speed was 10 mm min^−1^.

The tests were carried out in three consecutive steps: a pre-scan at 300 mN to determine the initial profile of the samples, a progressive scan from 0 to 25 N during which the penetration depth (P_d_) was recorded in real time, and a post-scan at 100 mN. SEM was used to study the shape of the residual deformations after the scratch tests. Finally, the indenter conducted a post-scan at a constant load of 300 mN and measured the residual depth (Rd) of the scratch. After the test, the scratch morphology was examined by SEM. At least three progressive load scratch tests were carried out for each sample.

### 2.6. Tribological Characterization

Tribological tests were performed at a constant normal load using a multi-pass scratch test (CSM Instruments, Anton Paar, Graz, Austria) under a unidirectional loading over a sliding distance (d) of 3 mm. Each test was carried out at a temperature of 22 °C and a relative humidity of 46%. The applied load, F_n_ (1 and 3 N), the number of passes (100 for each load condition), and the sliding speed (V) were controlled and fixed during the wear testing. Then, wear tracks were examined by SEM.

For each test, the wear volume (V) was calculated by measuring the surface profile on the wear track, and the dissipated energy was determined with the following Formula (1) [28].
(1)$$E=\sum \vec{F_{t}}\;\cdot \vec{d}=v\cdot t\;\sum F_{t}$$
where *F_t_* is the tangential load (N), *d* denotes the sliding distance (m), v is the sliding velocity (m/s), and *t* is the experiment duration (s). The wear coefficient is defined as the slope of the curve of the wear volume (V) vs. energy (*E*). A low coefficient wear corresponds to a high wear resistance.

### 2.7. Corrosion Test

The electrochemical experiments were performed in a conventional three-electrode system in which a saturated calomel electrode (SCE) was used as a reference electrode, a platinum sheet was taken as a counter electrode, and specimens (SS304 and FEP/SS304) with a 1.0 cm^2^ area were exposed as the working electrode. Electrochemical measurements were carried out in 3.5% NaCl solution after a stabilization period of 1 h to attain stable E_corr_ values. The experiments were also conducted on the coated samples after continuous immersion in 3.5 wt% NaCl solution for 60 days.

Polarization studies were performed in an Electrochemical VoltaLab PGZ 301 (Lyon, France). Both cathodic and anodic polarization curves were recorded, and Tafel polarization curves were obtained by varying the electrode potential value from −0.6 to 0.4 V at a scan rate of 10 mV S^−1^. The corrosion parameters included the corrosion potential (E_corr_) and the corrosion current (I_corr_), and the inhibition efficiency (I.E) was calculated by using Equation (2):(2)$$I.E=\left(1-\frac{{\mathrm{icorr}}_{coating}}{{\mathrm{icorr}}_{\mathrm{SS}304}\;}\right)\times 100$$

## 3. Results and Discussion

### 3.1. Microstructural Properties

The microstructure of the sprayed coatings is presented in Figure 2. From the top-view SEM micrographs (Figure 2a), it can be observed that the FEP coatings had a dense, poreless, and homogeneous structure. According to the EDS analysis (Figure 2b), the elementary composition of the prepared FEP coating consisted of C and F, suggesting that the coating was extraordinarily pure.

XRD spectrums of as-received FEP, FEP coating, and SS304 substrate are shown in Figure 2c. The XRD diffractogram of as-received FEP resin was characterized by a sharp peak with the highest intensity and a broad peak. In general, a polymer with a crystalline region presents sharp X-ray diffraction peaks, whereas the X-ray diffraction peaks are broad for an amorphous polymer [29]. The FEP resin had a semicrystalline structure which resulted in a definite crystalline peak at a value of 2θ = 17.85° and a spread amorphous peak at 2θ = 40°. Similar results were obtained by Tcherdyntsev et al. [30]. The XRD spectrum of the FEP coating showed that the crystalline peak in the 2θ range of 17–19° corresponded to the (100) plane of two-dimensional hexagonal packing [31]. According to Wesley et al. [31], the peak with the highest intensity (2θ = 17.85°) is generally associated with the crystalline phase, in which the incident X-ray is diffracted by the plane (100) of the pseudohexagonal lattice structure.

### 3.2. Mechanical Properties

To investigate the mechanical behavior of the FEP coatings, they were subjected to nanoindentation measurements. During the indentation test, the penetration depth increased as a function of the applied load, and a plastic deformation occurred until the maximum load corresponded to the maximum loading. A typical load–depth curve is plotted in Figure 3, and as can be seen, the FEP coatings demonstrated an elasto-plastic behavior. The analysis of the load–depth curves using the Oliver–Pharr method [27] makes it possible to estimate mechanical properties, i.e., hardness, H, and Young’s modulus, E, of films. The elastic modulus and the hardness of the coatings are presented in Figure 3b and were found to be 57 ± 2.35 MPa and 1.56 ± 0.07 GPa, respectively.

The indentation loading/unloading curve of the polymers appeared to be dependent on the holding time, as shown in the inset of Figure 3a. At the applied load of 10 mN, the penetration dept increased with time, indicating creep of FEP. The recovery index and indentation creep were derived from the standard nanoindentation measurements (Figure 3b). The film recovery index W_e_/W_t_ implied an elastic recovery during unloading and a significant plastic deformation (70%) during loading.

### 3.3. Scratch Behavior

Figure 4a–e shows the results of conventional scratch tests on the FEP coatings. As the normal load increased, the width of the scratch became larger, and partial delamination of the FEP coating occurred at the end of the groove. The scratch progression was accompanied by successive degradations defined by three different critical loads (L_c1_, L_c2_, L_c3_) [25]. These critical loads were determined by combining SEM observations of scratch tracks and the measurements of normal and tangential loads, as well as the penetration and residual depths during the scratch test.

The first critical load L_c1_ corresponding to the appearance of the first crack along the scratch pattern was localized at the edge, thus indicating cohesive failure. The second critical load L_c2_ was the applied normal load at which the extent of fracture events increased both at the bottom and at the edge of the scratch pattern, thereby leading to the observed repeated tensile cracking (adhesive failure). The third critical load L_c3_ was the applied normal load at which the coating exhibited a partial spalling.

Along the scratch path, the photos (Figure 4a–d) revealed that the scratch formed on the FEP coating could be divided into three stages. Stage I (until L_c1_ = 3.36 N), termed as smooth sliding, was the initial deformation stage of the FEP coating. Stage II (between L_c1_ = 3.36 N and L_c2_ = 6.2 N) was the transition region between smooth sliding and material removal featured by periodic micro-cracks. Stage III (from L_c2_ = 6.2 N) was defined as the material removal region where damage occurred and finally led to mass loss of FEP at L_c3_ = 7.6 N.

Figure 4e depicts the trends of the tangential force, penetration, and residual depth vs. normal force during the scratch test. The results were superimposed with the related SEM images of the residual scratch tracks of the investigated coatings. A purely elastic deformation took place at low applied scratch loads, and no scratch marks were detected along the starting segment of the scratch track, as shown in Figure 4a,e. By augmenting the normal load, as demonstrated in Figure 4e, a sudden slope change in the friction load and depth curves related to a first cracking phenomena was detected. This cohesive failure at L_c1_ was confirmed by SEM images (Figure 4b). When the applied load increased, the track demonstrated a propagation of micro-cracks, indicating adhesive failure at L_c2_. At the highest critical load L_c3_, a partial delamination was observed, thus indicating that the FEP coating bestowed good scratch resistance on the stainless-steel substrate.

### 3.4. Friction and Wear Resistance

Multi-pass sliding scratch testing was performed on the FEP coatings to evaluate the friction behavior and wear resistance. The SEM micrographs and EDS analyses of the multi-pass scratch test of FEP coatings at different applied loads are presented in Figure 5. The wear tracks obtained under 100 sliding cycles at constant loads of 0.3, 1, 2, and 3 N were examined using SEM. The applied load was lower than the cohesive failure load (L_c1_) of the coatings.

Figure 5 illustrates the SEM images of the scratch tracks at various applied loads (0.3, 1, 2, and 3 N). As demonstrated in Figure 5, after a 100-pass scratching, the width of the scratch tracks increased considerably as the loading was raised. At 0.3 N, the SEM results revealed plastic deformation located at the edge and a ductile ploughing in the center of the track. At 1 N, an increased plastic deformation was noticed at the edge and in the center of the tracks, but the film still adhered to its substrate, and only cohesive failure occurred. However, at higher loads (2 and 3 N), an extensive plastic deformation occurred, leading to a chipping phenomenon. At this stage, the FEP coating demonstrated delamination, which was an indication of adhesive wear.

This result was proved by EDS analysis of the wear tracks, as shown in Figure 5 and Table 2. For the applied load of 0.3 N, the chemical composition present in the wear track (Figure 5a’) was the same as that obtained in the film (see Figure 2b). Then, at 1 N (Figure 5b’), trace amounts of iron were noticed, but nevertheless, after 100 passes and at a load of 1 N, the film behavior remained cohesive.

However, at higher loads (2 and 3 N) (cf. Figure 5c,d’), the EDS analysis showed the appearance of the substrate elements C and Fe in the wear tracks during sliding (see Table 2). The elevated loading level and the high mechanical properties led to excessive Hertzian stresses, which promoted coating wear in the middle of the wear track [32].

The results of the wear tests in terms of friction coefficient and wear volume are illustrated in Figure 6. The evolution of the coefficients of friction of the coatings is presented as a function of the number of cycles for various applied loads in Figure 6a and after 100 cycles at various applied loads and velocities in Figure 6b.

The friction coefficient (COF) was recorded for 100 passes at each load. It can be noticed that the increase in the applied load led to a higher average value of the friction coefficient. At the lowest loads (0.3 and 1 N), the film had an exceedingly low friction coefficient that did not exceed 0.13 at 100 passes.

At higher applied loads (2 N and 3 N), the COF was, respectively, 0.18 and 0.24 at 100 passes. The increase in COF value for high applied loads was explained by the appearance of substrate after partial delamination of the FEP coating. The low friction coefficient reflects the high friction resistance of this coating. Such results were like those found by Nemati et al. [33]. Authors found that coating of the fluoropolymer PTFE on stainless steel was effective in lowering the COF from 1.2 to 0.16. The low and stable behavior of the COF was due to the synergistic lubrication effects of PTFE [34].

For each applied load and sliding velocity, the wear volume (V) and dissipated energy (E) were measured. The evolution of the wear volume was plotted versus the applied load for each sliding velocity, as shown in Figure 6c. Raising the applied load led to a significant increase in wear volume and dissipated energy regardless of the sliding velocity. This tendency was confirmed by the analysis of micrographs and the chemical composition of wear tracks, as previously observed. Concerning the velocity effect, the wear volume increased as the sliding velocity increased from 50 to 100 mm min^−1^.

The evolution of the wear volume (V) was plotted versus the dissipated energy (E) (see Figure 6d). Each point on this plot represents an experiment carried out at a given load (0.3, 1 N, 2 N, and 3 N) and for a given sliding speed (10, 50, and 100 mm min^−1^). The obtained graph reports a linear relation, and the slope of (E) vs. (V) is defined as the wear coefficient. Such an approach has already been successfully used in other works [26], and herein, the wear coefficient was equal to 3.12 × 10^−4^ mm^3^ N m^−1^. This was of the same order of magnitude as that reported in other studies using both PFA and PTFE with wear rates corresponding to ~10^−4^ mm^3^ N m^−1^ [35].

### 3.5. Corrosion Resistance

The cathodic and anodic polarization curves recorded for the SS304 and FEP coating at t = 1 h and after 60 days of immersion in 3.5 wt% NaCl solution are shown in Figure 7, and the specific data (Table 3) were obtained by Tafel fitting of these curves. The corrosion resistance of the coating can be described in detail by the electrodynamic polarization curve, and the lower polarization current indicated a superior corrosion resistance [36].

The extrapolation of the polarization curves was carried out to determine I_corr_ and E_corr_ (Figure 7, Table 3). As seen in Figure 7, the corrosion current density of SS304 decreased from 33 × 10^−4^ to 1.58 × 10^−4^ µA/cm^2^ with the coating deposition. In addition, the corrosion potential was raised from −450 mV to −240 mV. The enhancement of the SS304 corrosion resistance is associated with the physical-chemical characteristics of the FEP resins [10,37,38,39,40].

To further demonstrate the long-term anti-corrosion performance of the FEP coatings, polarization measurements were performed after immersing them in a 3.5 wt% NaCl corrosive environment for 60 days. According to Table 3 and Figure 7, a very small negative shift in E_corr_ and few positive shifts of I_corr_ were observed. This indicated that the anti-corrosion performance of the coatings was consistent after the immersion test. Moreover, the inhibition efficiency of the coatings to SS304 remained remarkable after the 60-day immersion test. The FEP coatings could further extend their protective effect by inhibiting the penetration of the corrosion medium into the stainless-steel substrate when in an aggressive and humid environment. Thus, FEP protected the product from any kind of chemical corrosion.

## 4. Conclusions

In this work, a protective FEP coating was deposited on a stainless-steel substrate using the air spray process and then cured to obtain a compact and uniform film. The mechanical, adhesion, tribological, and corrosion performance of FEP coatings were investigated by nanoindentation, scratch test, SEM, and cyclic voltammetry. The following conclusions were drawn from the experimental results. The FEP coatings exhibited a dense, poreless, and homogeneous structure with a pseudohexagonal lattice crystalline structure. They showed good mechanicals properties, hardness, and Young’s modulus and good scratch resistance to adhesive and cohesive failure, with a high adhesion to SS304. During the multi-pass scratch test of the FEP coating, both the wear volume and the dissipated energy increased when the normal applied load and sliding velocity were raised. It was also deduced that the coating exhibited a ductile behavior. After the wear tests, the wear on the FEP coatings with the increase of the applied load passed from cohesive wear to adhesive wear. The multi-pass scratch provided an easy and quick approach to study FEP wear resistance. The friction coefficient of the FEP coating did not exceed 0.13, and the wear coefficient was around 3.12 × 10^−4^ mm^3^ N m^−1^. The FEP coating enhanced the corrosion resistance of SS304 and provided a significant protection during 60 days of immersion in a NaCl solution.

## Figures and Tables

**Figure 1 polymers-14-00347-f001:**
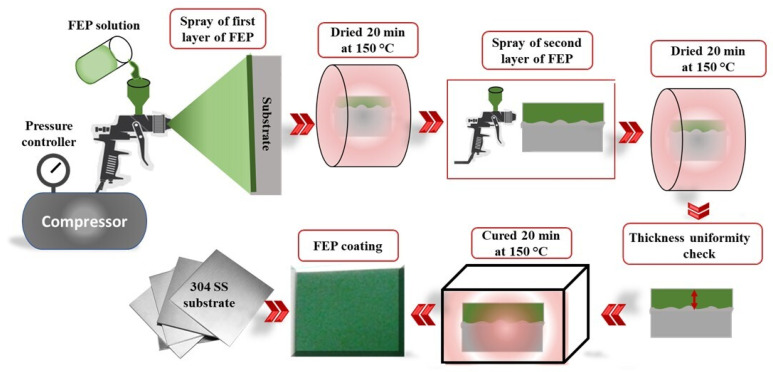
Processing cycle of FEP spraying on stainless steel.

**Figure 2 polymers-14-00347-f002:**
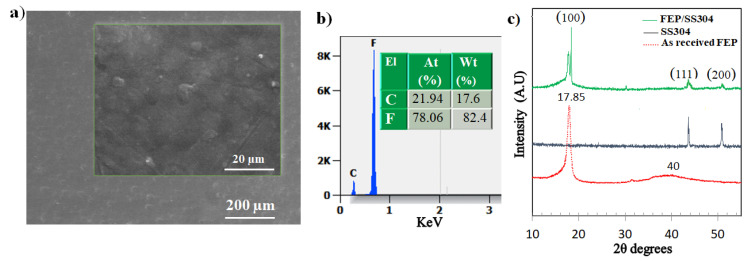
(**a**) SEM analysis of the FEP coatings, (**b**) a typical EDS spectrum, and (**c**) XRD pattern of the coatings.

**Figure 3 polymers-14-00347-f003:**
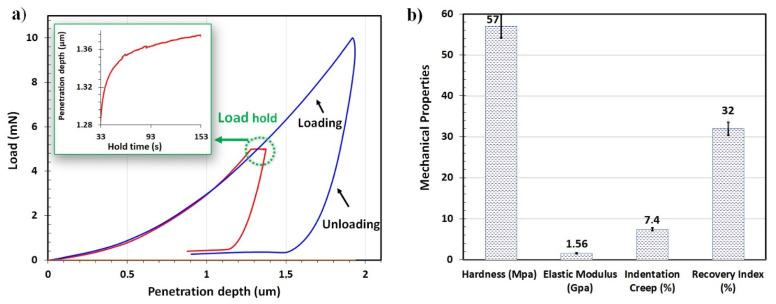
**(****a**) Load–depth curves for FEP coatings with 5 and 10 m N maximum load and (**b**) measured values from nanoindentation tests on FEP coatings.

**Figure 4 polymers-14-00347-f004:**
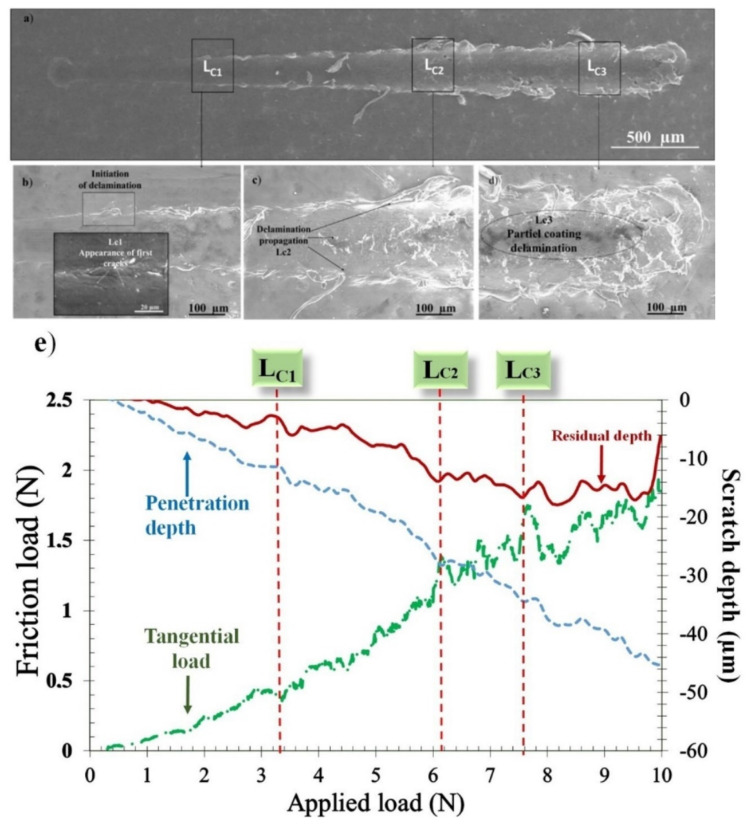
Progressive scratch test on an FEP coating. (**a**) SEM micrographs of scratch tracks. Magnified photo at (**b**) L_c1_, (**c**) L_c2_, and (**d**) L_c3_. (**e**) Tangential load, penetration, and residual depths as functions of the applied load.

**Figure 5 polymers-14-00347-f005:**
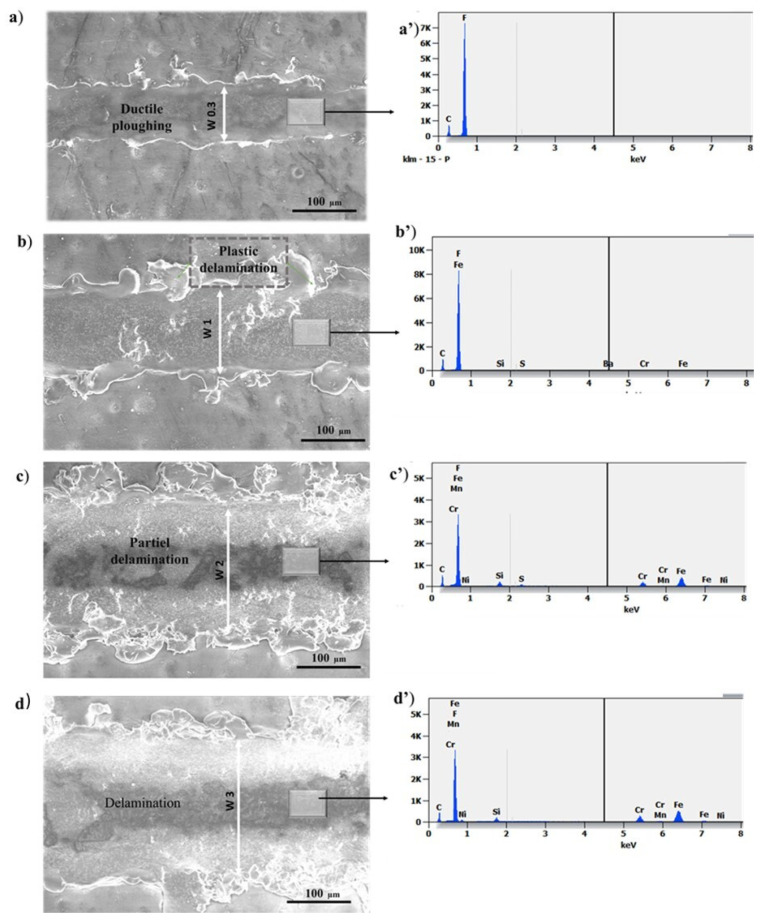
SEM micrographs and corresponding EDS analyses of 100-pass scratch tracks at (**a**) 0.3 N, (**b**) 1 N, (**c**) 2 N, and (**d**) 3 N.

**Figure 6 polymers-14-00347-f006:**
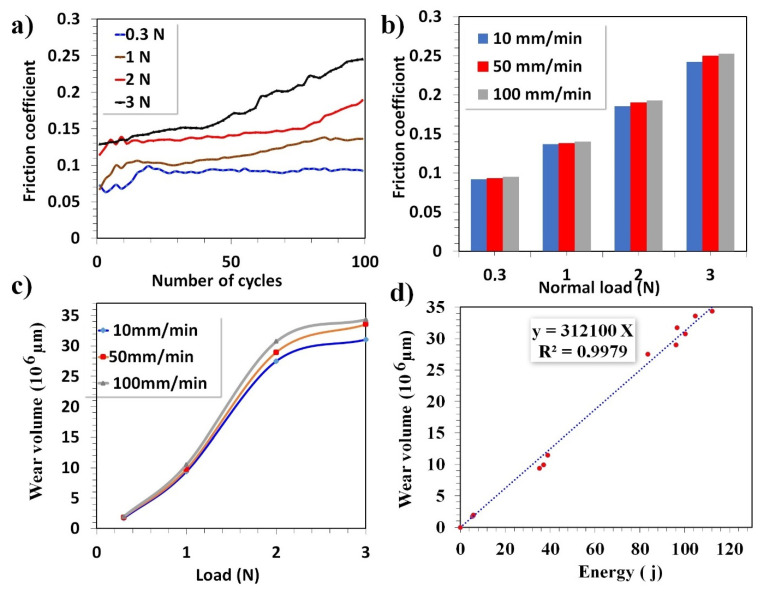
(**a**) COF plot of FEP coatings versus the number of cycles for 0.3, 1, 2, and 3 N at 50 mm min^−1^, (**b**) COF bar graph and (**c**) wear volume versus load after 100 passes at various velocities, and (**d**) wear volume as a function of the dissipated energy.

**Figure 7 polymers-14-00347-f007:**
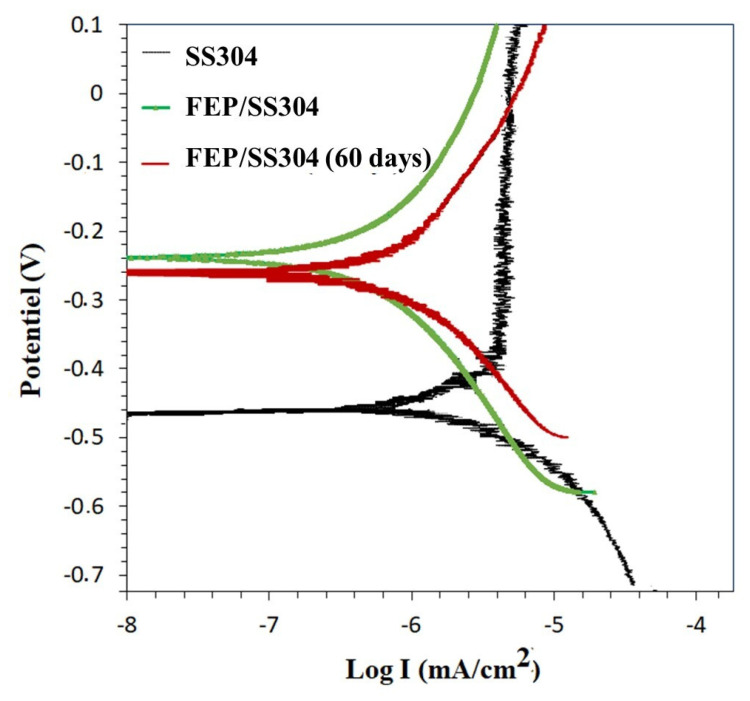
Polarization curve of SS304 and FEP coatings in 3.5% NaCl for 1 h and for 60 days.

**Table 1 polymers-14-00347-t001:** Chemical composition of SS304.

Element	Fe	C	Si	Mn	P	S	N	Cr	Mo	Ni	Cu	Co
W_t_ (%)	70.594	0.049	0.42	1.9	0.024	0.025	0.078	18.1	0.35	8.06	0.35	0.05

**Table 2 polymers-14-00347-t002:** Chemical composition of the film after and before wear tracks.

	Chemical Composition (wt%)							
	C	F	Si	S	Mn	Cr	Fe	Ni
Track of 0.3 N	21.10	78.90						
Track of 1 N	20.09	77.62	0.125	0.015		0.40	1.75	
Track of 2 N	14.45	63.56	0.80	0.020	0.38	4.90	14.3	1.59
Track of 3 N	10.58	54.6	0.86	0.02	0.46	6.86	24.17	2.45

**Table 3 polymers-14-00347-t003:** Potentiodynamic polarization data of SS304 and FEP coating.

Immersion Time	Samples	E_corr_ (mV)	I_corr_ (µA/cm^−2^)	Inhibition Efficiency (%)
1 h	SS304	−450	33.1 × 10^−4^	-
	FEP/SS304	−240	1.58 × 10^−4^	95.22
60 days	FEP/SS304	−264	2.51 × 10^−4^	92.4

## Data Availability

Not applicable.

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
