# Peer review of "Fluorinated Ethylene Propylene Coatings Deposited by a Spray Process: Mechanical Properties, Scratch and Wear Behavior"

_polymers, 2022, doi:10.3390/polym14020347_

Round 1

Reviewer 1 Report

The review refers to the manuscript titled "Fluorinated Ethylene Propylene Coatings deposited by a Spray Process: Mechanical properties, Scratch and Wear behavior".

In the section "Introduction", please describe what the cited publications present, not just add the numbers of items (e.g. [8-12]) from the list of references.

Please check the correctness of the entries, because there are typos, e.g. x-ray instead of X-ray, young's modulus instead of Young's modulus, etc.

Please improve the quality of the photos and the readable of the drawings.

Axis descriptions in Figure 3 (b) are missing.

How many repetitions of the tests described in paragraph 2.2 were carried out?
The figures do not show that the tests were repeated, because the results were given as if for one example of each test. Please explain this and complete it.

Author Response

Response to Reviewer 1 Comments

Authors are grateful to the reviewer for the comments. Moreover, suggestions, comments and questions of the reviewer are useful a point-by-point response.

Point 1: In the section "Introduction", please describe what the cited publications present, not just add the numbers of items (e.g. [8-12]) from the list of references.

Response 1: We agree with the reviewer, as suggested, we have described what the cited publication present in the revised version.

Point 2: Please check the correctness of the entries, because there are typos, e.g. x-ray instead of X-ray, young's modulus instead of Young's modulus, etc.

Response 2: Thank you for pointing this typing error to us. The error was corrected in the revised version of the paper.

Point 3: Please improve the quality of the photos and the readable of the drawings.

Response 3: Thanks to the reviewer for careful assessments. The quality of the photos was improved.

Point 4: Axis descriptions in Figure 3 (b) are missing.

Response 4: We absolutely agree with the reviewer. In our revised paper, axis descriptions were added in Figure 3.

Point 5: How many repetitions of the tests described in paragraph 2.2 were carried out?
The figures do not show that the tests were repeated, because the results were given as if for one example of each test. Please explain this and complete it.

Response 5: Thank you for the reviewer comment. All the measurements provided in the tests have been carried on 5 times, this information was added in the revised paper. The figure 3 show that the tests were repeated (see error bars). In addition, our study was based on repeated experiments (five times) and results were reproducible. This was explained by the homogeneous and uniform films that were deposited in RIET industry (specialist in industrial coatings). This explanation was added int the revised text (see material and method section).

In summary, we greatly appreciate the reviewer for the valuable suggestions. The paper has been carefully checked and fully revised. All the concerns have been thoroughly taken into consideration in the revision. The English language is also enhanced. We would gladly address any other concerns if the reviewer deems necessary and we hope that the Reviewer will be satisfied with our responses. We are also pleased to resubmit for publication the revised version of our manuscript and we look forward to your positive response.

Sincerely

Reviewer 2 Report

Barhoumi et al. presented work regarding FEP coating on stainless steel and studied its mechanical, scratch, and wear behaviour. The results are satisfactory but the authors need so much improvement throughout the manuscript especially the introduction and materials and methods section. I do not find any discussion part in the manuscript. Additional authors should do add some new experiment parts (AFM (for surface topography and roughness) and white light interferometry (WHL) (for thickness measurement)).

Comments

  1. Figure 1 should be renamed as scheme: 1; accordingly change numbering in all figures.

In figure 1, Dried “20 mn”, please correct it as min in two times, Line 107, second round or second layer. Make it consistent in the whole manuscript as well as the figure. Caption of Figure 1, Line 104, ‘….FEP spraying process” Please write it as “…FEP coating process”.

  1. Authors coding as stainless steel as SS304 but authors have written as 304 SS in many places in text, tables and figures, so please make it consistent throughout the manuscript inside the text, figures, and tables.
  2. Section 3.1, there is no discussion part. Please improve it. Line 175, Is it Results only or both? Please mention it clearly. Authors should insert the JCPDS file no in XRD figure and make also (hkl) plane inside the figure. In Fig. 2c, in the x-axis, Major and minor ticks are missing. Mention it clearly. Mention the crystallinity in the result section. Follow this manuscript to make your Xrd figure in the right way and cite it. https://doi.org/10.1016/j.jlumin.2020.117593, https://doi.org/10.1016/j.carbpol.2021.117633. Please provide also  pure FEP Xrd pattern.

In section 3.2, the authors described the mechanical properties of the coating sample. Author should compare the mechanical properties of the pristine substrate and coated substrate.

  1. Line no 101, 2.5 mL/min, please rewrite it 2.5 mL min-1; check this type of mistake throughout the manuscript.
  2. Please correct all superscript and subscript notation throughout the manuscript.
  3. Cl- should be - should be clearly placed as a superscript. In line 15-17, put SEM abbreviation after scanning electron microscopy and EDS as well.
  4. In Figure 5, ‘d’ should in bracket as d).
  5. All the figures clarity should be enhanced. Minor and major ticks should be marked in x and y-axis improved as well.
  6. In figure 7, y-axis 0,4 and other labels, should be the point as 0.4. correct it throughout the manuscript.
  7. The conclusion should be one paragraph please avoid bulletin. 
  8. Cite some recent references (2021) in the results and discussion and introduction part.
  9. Cite a few references related to this study from MDPI "polymers" journal. Line 261, what is (?)
  10. The introduction is messy; authors should highlight the novelty briefly.
  11. References should be consistently placed for example Ref no 11 authors used journal abbreviations but other references authors used full names. 
  12. Ref 11: Application of thermal spraying in the automobile industry: all the letters are small, except A in Application but in some other references some time letters are big and sometimes small. make it consistent throughout the manuscript.
  13. References are very few in the manuscript please cite more papers make it at least more than 40.
  14. Provide a comparison table with your recent work. In Table 2, % ATOMIQUE, check its appropriate term.
  15. Table 1 is not necessary it's already analyzed by the company because it is the commercial substrate. Just in materials clearly mention where you bought. In section 2, please clearly first write 2.1 as ‘Materials’ and 2.2 and so on… Mention FEP concentration, how authors prepare FEP solutions. How authors optimized a particulate conc. attached on the substrate. Please explain it in the manuscript. Please mention instruments and chemicals name, the country which is used in this study
  16. Section 2.2.4 equation number is missing. make equation in normal font. Not bold.
  17. Line 171 I.E make it normal font. Line 134 the ‘T’ should be capitalized.
  18. Line 320, make space between number and mV.
  19. For AFM measurement, authors can follow this manuscript and cite the paper in https://doi.org/10.1016/j.ijbiomac.2020.12.226 and present their results accordingly.
  20. How authors measure the thickness of the coating part; coating amount is too small. It is impossible to measure by standard gauge; it should be measured by WHL instrument or AFM technique.
  21. The surface roughness should be measured by AFM.
  22. Line 66, “260°C and 360°C" there should be space between 260 and °C 
  23. Line 47, FEP, PTFE, and PFA please write their full name and then use their abbreviation throughout the manuscript.
  24. Line 189, Figure 2. (a) SEM analysis of the FEP coatings, (b) XRD pattern of the coatings. and (c) a typical EDS spectrum. remove. from the XRD pattern of the coatings. and put their comma.

Fig. 2 (c), 2 Theta (deg), should be rewritten as 2-theta (deg.), major-minor ticks should be marked clearly in the x-axis.

  1. The chemical structure of the coating onto SS3.4, authors should include the FTIR result and solid-state NMR result.
  2. The manuscript has 34% plagiarism now; please avoid it and make it less than 20% at least.
  3. Figure 6d, R2= 0,9979 please avoid comma and put decimal. Also in Line 28 (3,1). Also check this error throughout the manuscript in text, figures and tables.
  4. The authors should discuss the other polymers coating on SS304 steel and compare advantages and disadvantages in the introduction part as well as the results and discussion part.
  5. Please provide the Potentiodynamic polarization data at least every 10 days.
  6. In the conclusion part, Line 339, please make small C and V.
  7. English should be corrected throughout the manuscript.

Author Response

Response to Reviewer 2 Comments

Authors are grateful to the reviewer for the comments. Moreover, suggestions, comments and questions of the reviewer are useful a point by point response.

Point 1: Figure 1 should be renamed as scheme: 1; accordingly change numbering in all figures.

In figure 1, Dried “20 mn”, please correct it as min in two times, Line 107, second round or second layer. Make it consistent in the whole manuscript as well as the figure. Caption of Figure 1, Line 104, ‘….FEP spraying process” Please write it as “…FEP coating process”.

Response 1: Thanks for the reviewer comment. According to reference [9], published in MDPI, Polymers, the title of Figure 1 was changed to be “The processing cycle of FEP sprayed on stainless steels”. 

[9] Rodríguez-Alabanda, Ó.; Romero, P.E.; Soriano, C.; Sevilla, L.; Guerrero-Vaca, G. Study on the Main Influencing Factors in the Removal Process of Non-Stick Fluoropolymer Coatings Using Nd:YAG Laser. Polymers 2019, 11, 123. https://doi.org/10.3390/polym11010123

Point 2: Authors coding as stainless steel as SS304 but authors have written as 304 SS in many places in text, tables and figures, so please make it consistent throughout the manuscript inside the text, figures, and tables.

Response 2: We absolutely agree with the reviewer’s comment. In our revised paper, the stainless steel was coding as SS304 in all the text, tables, and figures. (figure 1 et corrosion

Point 3: Section 3.1, there is no discussion part. Please improve it. Line 175, Is it Results only or both? Please mention it clearly. Authors should insert the JCPDS file no in XRD figure and make also (hkl) plane inside the figure. In Fig. 2c, in the x-axis, Major and minor ticks are missing. Mention it clearly. Mention the crystallinity in the result section. Follow this manuscript to make your Xrd figure in the right way and cite it. https://doi.org/10.1016/j.jlumin.2020.117593, https://doi.org/10.1016/j.carbpol.2021.117633. Please provide also  pure FEP Xrd pattern.

In section 3.2, the authors described the mechanical properties of the coating sample. Author should compare the mechanical properties of the pristine substrate and coated substrate.

Response 3: We absolutely agree with the reviewer’s comment. In our revised paper, the discussion in Section 3.1 was improved and we added two references [ggggg].  In Line 175, the results show both cathodic and anodic polarization curves as showed in figure below.

in the submitted report 

For XRD result, we agree with the reviewer comment, (hkl) plane and the major and minor ticks are added in figure 2c. Drawing the viewer's attention to the purity of FEP that was 99.99. on the other hand, we did not measure the mechanical properties of the substrate because our objective is to deposit a protective coating against corrosion and wear (improvement of  the molds life).

Point 4: Line no 101, 2.5 mL/min, please rewrite it 2.5 mL min-1; check this type of mistake throughout the manuscript.

Response 4: Thank you for pointing this typing error to us. The error was corrected in the revised version of the paper. 

Point 5: Please correct all superscript and subscript notation throughout the manuscript.

Response 5: Thanks to the reviewer for the comment. all superscript and subscript notation was corrected throughout the manuscript.

Point 6: Cl- should be - should be clearly placed as a superscript. In line 15-17, put SEM abbreviation after scanning electron microscopy and EDS as well.

Response 6: Thanks to the reviewer for the comment. Cl- should was placed as a superscript. SEM and EDS abbreviation were added as suggested.

Point 7: In Figure 5, ‘d’ should in bracket as d).

Response 7: In Figure 5, ‘d’ was changed in bracket as d).

Point 8: All the figures clarity should be enhanced. Minor and major ticks should be marked in x and y-axis improved as well.

Response 8: all figures clarity was enhanced. Minor and major ticks were marked as suggested.

Point 9: In figure 7, y-axis 0,4 and other labels, should be the point as 0.4. correct it throughout the manuscript.

Response 9: In figure 7, y-axis 0,4 and other labels, were corrected it throughout the manuscript.

Point 10: The conclusion should be one paragraph please avoid bulletin. 

Response 10: We have considered in our revised paper, the reviewer comment and the conclusion were changed in one paragraph. 

Point 11: Cite some recent references (2021) in the results and discussion and introduction part.

Response 11: as suggested, we have Cited some recent references (2021) in the results and discussion and introduction part.

Point 12: Cite a few references related to this study from MDPI "polymers" journal. Line 261, what is (?)

Response 12: As suggested, references related to this study from MDPI "polymers" journal was added. For Line 261, (?) is a typing error that was corrected.

Point 13: The introduction is messy; authors should highlight the novelty briefly.

Response 13: We agree with this suggestion. In the introduction of revised version, we highlight the novelty briefly.

Point 14: References should be consistently placed for example Ref no 11 authors used journal abbreviations, but other references authors used full names. 

Response 14: All References were changed as suggested.  Ref no 11 was changed by “Primc, G. Recent Advances in Surface Activation of Polytetrafluoroethylene (PTFE) by Gaseous Plasma Treatments. Polymers 2020, 12, 2295. https://doi.org/10.3390/polym12102295”.

Point 15: Ref 11: Application of thermal spraying in the automobile industry: all the letters are small, except A in Application but in some other references some time letters are big and sometimes small. make it consistent throughout the manuscript.

Response 15: Ref 11 was modified.

Point 16: References are very few in the manuscript please cite more papers make it at least more than 40.

Response 16: References were changed to be  more than  40.

Point 17: Provide a comparison table with your recent work. In Table 2, % ATOMIQUE, check its appropriate term.

Response 17: Thanks to the reviewer’s comment. To the best of our knowledge, there is no study in the literature that investigated the scratch behavior of FEP coating. In Table 2, % ATOMIQUE was removed and substituted by Chemical composition (At%). %).

Point 18: Table 1 is not necessary it's already analyzed by the company because it is the commercial substrate. Just in materials clearly mention where you bought. In section 2, please clearly first write 2.1 as ‘Materials’ and 2.2 and so on… Mention FEP concentration, how authors prepare FEP solutions. How authors optimized a particulate conc. attached on the substrate. Please explain it in the manuscript. Please mention instruments and chemicals name, the country which is used in this study.

Response 18: Thanks to the reviewer for the comment. The table of commercial substrate was presented to take it a reference for the analysis of tracks of scratch (chemical composition).

In section 2, the write ( 2.1 as ‘Materials’ and 2.2 ..)were changed as suggested in In the revised version.

The FEP solution were an  aqueous solution supplied by Whitford England. The coating deposition was carried out on RIET industry -Tunisia.

The reviewer comments were considered and explained in the revised paper. In the revised version.

Point 19: Section 2.2.4 equation number is missing. make equation in normal font. Not bold.

Response 19: The reviewer comment was considered. In the revised version, equation number in Section 2.2.4 was added. 

Point 20: Line 171 I.E make it normal font. Line 134 the ‘T’ should be capitalized.

Response 20: Thanks to the reviewer for careful assessments. The text was changed as suggested. 

Point 21: Line 320, make space between number and mV.

Response 21: Text was changed as suggested.

Point 22: For AFM measurement, authors can follow this manuscript and cite the paper in https://doi.org/10.1016/j.ijbiomac.2020.12.226 and present their results accordingly. 

Response 22: In our study there are no AFM measurement. But the substrate roughness (before the deposition of FEP coating) was measured and added in the text.

Point 23: How authors measure the thickness of the coating part; coating amount is too small. It is impossible to measure by standard gauges; it should be measured by WHL instrument or AFM technique.

Response 23: Thanks for the reviewer comment. The thickness of the coating was measured using an Ecotest plus device gauge. (this information was mentioned in the text). The measurements were carried out in RIET industry. The coatings thickness was measured at about 44 µm ± 4 ( is not too small).

Point 24: The surface roughness should be measured by AFM. 

Response 24: Thanks for the reviewer comment. The surface roughness of substrate was added the manuscript and were measured by a rugosimeter.

Point 25: Line 66, “260°C and 360°C" there should be space between 260 and °C

Response 25: Thanks to the reviewer for the comment, text was corrected as suggested.

Point 26: Line 47, FEP, PTFE, and PFA please write their full name and then use their abbreviation throughout the manuscript.

Response 26: Thanks to the reviewer for the comment, In the revised version, the full name of FEP, PTFE and PFA are added and then we have used their abbreviation throughout the manuscript.

Point 27: Line 189, Figure 2. (a) SEM analysis of the FEP coatings, (b) XRD pattern of the coatings. and (c) a typical EDS spectrum. remove. from the XRD pattern of the coatings. and put their comma.

Fig. 2 (c), 2 Theta (deg), should be rewritten as 2-theta (deg.), major-minor ticks should be marked clearly in the x-axis.

Response 27: Thanks for the reviewer comment. Figure 2. (a) SEM analysis of the FEP coatings, (b) XRD pattern of the coatings. and (c) a typical EDS spectrum were modified as suggested. Then, Fig. 2 (c), 2 Theta (deg), was changed to be 2-theta (deg.) and the major-minor ticks were marked clearly in the x-axis. 

Point 28: The chemical structure of the coating onto SS3.4, authors should include the FTIR result and solid-state NMR result. 

Response 28:  We would like to thank the reviewer for his comment. FTIR and solid-state NMR result are interesting. In our study we hope that we have well exploited the MEB and DRX to investigate the FEP coating microstructure.  Then, in a mechanical laboratory it is no obvious to have FTIR and solid-state NMR techniques.

Point 29: The manuscript has 34% now; please avoid it and make it less than 20% at least.

Response 29:  Thanks for the reviewer comment. In our revised paper, the discussion was improved, and the plagiarism was avoided.  

Point 30: Figure 6d, R2= 0,9979 please avoid comma and put decimal. Also in Line 28 (3,1). Also check this error throughout the manuscript in text, figures and tables.

Response 30:  Thanks for the reviewer comment. Figure 6d was changed as suggested. We have checked that there are no further errors (comma) throughout the manuscript in text, figures and tables. 

Point 31: The authors should discuss the other polymers coating on SS304 steel and compare advantages and disadvantages in the introduction part as well as the results and discussion part.

Response 31: We absolutely agree with the reviewer’s comment. In our revised paper, the discussion in Section 3.1 was improved and we added two references 

Nemati, N., Emamy, M., Yau, S., Kim, J. K., & Kim, D. E. (2016). High temperature friction and wear properties of graphene oxide/polytetrafluoroethylene composite coatings deposited on stainless steel. RSC advances, 6(7), 5977-5987. 

https://doi.org/10.1039/C5RA23509J

Wang, H.; Sun, A.; Qi, X.; Dong, Y.; Fan, B. Experimental and Analytical Investigations on Tribological Properties of PTFE/AP Composites. Polymers 2021, 13, 4295. https://doi.org/10.3390/polym13244295

Point 32: Please provide the Potentiodynamic polarization data at least every 10 days. No corrosion in 60 day.

Response 32: Thanks for the reviewer comment.  The corrosion test was repeated several times and we have noticed that no corrosion was found after 60 days, for this reason we believe that it is not important to test the corrosion resistance after only 10 days.

Point 33: In the conclusion part, Line 339, please make small C and V.

Response 33: Thanks for the reviewer comment. Conclusion part was changed as suggested. 

Point 34: English should be corrected throughout the manuscript.  

 Response 34: Thanks for the reviewer comment. According to the comment, English was improved in the revised manuscript

 Sincerely

Round 2

Reviewer 1 Report

Two more comments
1. The photos could be of better quality.
2. In the section "Introduction", there are still the numbers of items (e.g. [8-12], [23-25], etc.), please add description what the cited publications present.

Author Response

Response to Reviewer 1 Comments

Authors are grateful to the reviewer for the comments. Moreover, suggestions, comments and questions of the reviewer are useful a point-by-point response.

Point 1: The photos could be of better quality.

Response 1: Thanks to the reviewer for careful assessments. The quality of the photos was improved.

Point 2: In the section "Introduction", there are still the numbers of items (e.g. [8-12], [23-25], etc.), please add description what the cited publications present.

Response 2: We agree with the reviewer, as suggested, we have described what the cited publication present in the revised version. And for 23-25 the description added as suggested.

For the cited publications 8 and 9 and 10 the description are added in the later revised manuscript

(Nickel based Metallic and Zirconia based ceramic protective coatings were deposited on Steel diecasting molds [8]. According to Óscar Rodríguez-Alabanda [9] the fluoropolymer coatings are applied as anti-adherent coatings on aluminum–magnesium substrates for food containers. In our previous work [10], a polymeric perfluoroalkoxy coatings were used to protect agri-food molds to prevent corrosion and wear)

In summary, we greatly appreciate the reviewer for the valuable suggestions. and we hope that the Reviewer will be satisfied with our responses. We are also pleased to resubmit for publication the revised version of our manuscript and we look forward to your positive response.

Sincerely

Reviewer 2 Report

Authors improved the manuscript. However, still, there are many issues that need to be carefully revised.

-In Figure 1, 20mn, should be changed to ‘20 min’. Please check it carefully.

-In Figure 7, caption inside authors 304SS, Line 348, SS304, please check carefully throughout the manuscript to make consistency. Check-in table 3.

-Equation 2, Icorr ‘coaing’ need to be corrected as ‘coating’.

-Section 3, should be written as ‘Results and discussion’

-Line 200, Make it is clear and rewrite this sentence, “This result was recorded by Wesley Lock Sulen [30]”.

-Authors did not indicate XRD planes of all the peaks. Please provide JCPDS data pdf number to support your results if you don’t want to give XRD pattern of FEP. Otherwise it will be misunderstanding for readers. Please follow this and cite the paper to calculate the crystallinity of the coating and pristine substrate. https://doi.org/10.1016/j.jlumin.2020.117593.  

-Authors have not checked all the errors of Revision 1 in point no 4, please check inside the figure also for example in Figure 6. mm/min should mm min-1. Please check such types of error throughout the manuscript carefully.

-In revision 1, point 22, the authors have not measured the roughness of the samples. It is not in the revised manuscript.

-Still have manuscript showing 31% plagiarism, many sentences directly copied from other published papers. Kindly avoid it.

-Section 2.1, should be only materials

Author Response

Response to Reviewer 2 Comments

We highly appreciate the detailed and valuable comments from the reviewers regarding our manuscript. In this revised draft, all the reviewer’ comments have been addressed. A point-by- has been appended to this letter..  We would like to thank the Editor, and the reviewers for their careful assessments and constructive comments on our submission, particularly the time spent.

Point 1:

Authors improved the manuscript. However, still, there are many issues that need to be carefully revised.

-In Figure 1, 20mn, should be changed to ‘20 min’. Please check it carefully.

Response 1: We thank the reviewer for their careful assessments. The figure has been edited as suggested in the revised text.

Point 2:

-In Figure 7, caption inside authors 304SS, Line 348, SS304, please check carefully throughout the manuscript to make consistency. Check-in table 3.

Response 2 :

We absolutely agree with the reviewer’s comment. In our revised paper, the stainless steel was coding as SS304 in all the text, table3, and figure 7

Point 3:

-Equation 2, Icorr ‘coaing’ need to be corrected as ‘coating’.

Response 3:

Thanks to the reviewer for the comment. The error was corrected in the revised version as ‘coating’

Point 4:

-Section 3, should be written as ‘Results and discussion’

Response 4 :

Thanks to the reviewer for the comment. Section 3 was changed as suggested.

Point 5:

-Line 200, Make it is clear and rewrite this sentence, “This result was recorded by Wesley Lock Sulen [30]”.

Response 5 :

We agree with the reviewer, as suggested, we have rewritten the sentence in in the revised version of the paper.

Point 6:

-Authors did not indicate XRD planes of all the peaks. Please provide JCPDS data pdf number to support your results if you don’t want to give XRD pattern of FEP. Otherwise it will be misunderstanding for readers. Please follow this and cite the paper to calculate the crystallinity of the coating and pristine substrate. https://doi.org/10.1016/j.jlumin.2020.117593.  

Response 6 :

We absolutely agree with the reviewer’s comment.  In our revised paper, the XRD pattern of as received FEP was given. And the XRD pattern of coating was modify by another corrected XRD test on FEP coating.

In the previous XRD pattern of FEP coating, the peak of 35.5 corresponds to a silicon carbide which was used as blasting materials to prepare the coated substrate surface 5SS304)  .

In our revised paper, we added XRD pattern of FEP and the corrected XRD PATTERN of FEP coating as suggested.  (Since we have several tests of DRX we have selected the reproducible pattern).  

Based on the Study of WESLY et al and Tcherdyntsev et al we have edited the part of discussion of DRX.

[30] Tcherdyntsev, V. V., Olifirov, L. K., Kaloshkin, S. D., Zadorozhnyy, M. Y., & Danilov, V. D.. Thermal and mechanical properties of fluorinated ethylene propylene and polyphenylene sulfide-based composites obtained by high-energy ball milling. Journal of Materials Science, 2018, 53(19), 13701-13712. https://doi.org/10.1007/s10853-018-2508-9

[31] Wesley Lock Sulen, Kesavan Ravi, Chrystelle Bernard, Yuji Ichikawa, Kazuhiro Ogawa. Deposition Mechanism Analysis of Cold-Sprayed Fluoropolymer Coatings and Its Wettability Evaluation. Journal of Thermal Spray Technology, ASM International/Springer, 2020, 29 (7), pp.1643-1659. ff10.1007/s11666-020-01059-wff. ffhal-02970940.

Point 7: Authors have not checked all the errors of Revision 1 in point no 4, please check inside the figure also for example in Figure 6. mm/min should mm min-1. Please check such types of error throughout the manuscript carefully.

Response 7 :

Thank you for pointing this typing errors to us. The error was corrected in the revised version of the paper.

Point 8: In revision 1, point 22, the authors have not measured the roughness of the samples. It is not in the revised manuscript.

Response 8 :

We absolutely agree with the reviewer’s comment. we have not measured the roughness of the samples after coating but just the roughness of the substrate after preparation. the substrate roughness (before the deposition of FEP coating) was added in the text. (it was roughened by sandblasting to a surface finish Ra ≈ 3 µm).

Point 9: Still have manuscript showing 31% plagiarism, many sentences directly copied from other published papers. Kindly avoid it.

 Response 9: Thanks for the reviewer comment. In our revised paper, the text was edited and plagiarism was avoided. 

Point 10: Section 2.1, should be only materials

Response 10: We have considered in our revised paper, the Section 2.1, was changed to  materials.

In summary, we greatly appreciate the valuable suggestions from the reviewer. The paper has been carefully checked and fully revised. We would gladly address any other concerns that the reviewers deem necessary, and we hope that the reviewers and the editors will be satisfied with our responses. We are pleased to resubmit for publication the revised version of our manuscript and we look forward to your positive response.

Sincerely
